# Enhancement of RecET-mediated *in vivo* linear DNA assembly by a *xonA* mutation

James A. Sawitzke[1,2¤a]*, Nina Costantino[2,3¤a], Adriana Castillo Caballero[1], Ellen Hutchinson[2¤b], Alessandro Barenghi[1], Lynn C. Thomason[2,4¤c], Donald L. Court[2,4¤a]

**1** Genetic & Viral Engineering Facility, Epigenetics and Neurobiology Unit, European Molecular Biology Laboratory, Monterotondo (RM), Italy, **2** Gene Regulation and Chromosome Biology Laboratory, Frederick National Laboratory for Cancer Research, National Cancer Institute, Frederick, Maryland, United States of America, **3** RNA Biology Laboratory, Frederick National Laboratory for Cancer Research, National Cancer Institute, Frederick, Maryland, United States of America, **4** Basic Science Program, Frederick National Laboratory for Cancer Research, Frederick, Maryland, United States of America

¤a Current address: Frederick, Maryland, United States of America
¤b Current address: Ascension Street Vincent Women and Infants Hospital, Indianapolis, Indiana, United States of America
¤c Current address: Eugene, Oregon, United States of America
* jim.sawitzke@gmail.com

## Abstract

Assembly of intact, replicating plasmids from linear DNA fragments introduced into bacterial cells, i.e., *in vivo* cloning, is a facile genetic engineering technology that avoids many of the problems associated with standard *in vitro* cloning. Here, we report characterization of various parameters of *in vivo* linear DNA assembly mediated by either the RecET recombination system or the bacteriophage λ Red recombination system. As previously observed, RecET is superior to Red for this reaction when the terminal homology is 50 bases. Deletion of the *E. coli xonA* gene, encoding Exonuclease I, a 3'→5' single-strand DNA exonuclease, substantially improves the efficiency of *in vivo* linear DNA assembly for both systems. Deletion of the Exonuclease I function allows robust RecET assembly of six DNA segments to create a functional plasmid. The linear DNAs are joined accurately with very few errors. This activity is at least as efficient and accurate as the NEBuilder® HiFi DNA Assembly *in vitro* method of assembling fragments. This discovery provides a significant improvement to previously reported *in vivo* linear DNA assembly technologies and provides a faster, less expensive, one-step method for assembling plasmids from multiple fragments.

## Introduction

Recombineering is an *in vivo* genetic engineering technique that utilizes bacteriophage-derived recombination proteins to introduce precise genetic modifications into bacterial chromosomes or episomes. These proteins are able to utilize short

**Data availability statement:** All relevant data are within the paper and its Supporting Information files.

**Funding:** This work was supported, in part, by the Intramural Research Program of the National Institutes of Health, National Cancer Institute, Center for Cancer Research through D.L.C. This project has also been partly funded with federal funds from the National Cancer Institute, National Institutes of Health, under contract no. HHSN261200800001E through D.L.C. Funding was provided by the European Molecular Biology Laboratory (EMBL) to J.A.S.

**Competing interests:** No authors have competing interests.

(~50 base) homologies with double-stranded (dsDNA) [1,2] or single-stranded (ssDNA) [3] linear DNA acting as substrates. Such phage recombination systems typically include two proteins that are co-expressed and thus act coordinately [4]. A 5'→3' dsDNA-dependent exonuclease [5] processes linear dsDNA to leave single-strand 3' overhangs; these overhangs are bound by a single-strand annealing protein (SSAP) [6]. This SSAP-ssDNA complex is annealed to its complementary single-strand target sequence in the bacterial cell. Both proteins are required for recombination of dsDNA [1,2,6,7], while only the SSAP is required for ssDNA recombination [3]. For phage λ, these proteins are known as the Red system: the exonuclease is Red Exo (226aa) [5], and the SSAP, Red Beta (261aa) [6–8]. A second commonly used recombineering system, RecET, is derived from the *E. coli* cryptic prophage Rac [9]. RecE (866aa) is the 5'→3' dsDNA-dependent exonuclease [10]; however, it is nearly four times the size of λ Exo, with the exonuclease domain in the last ~260aa [10,11]. The function of the large, dispensable RecE N-terminus is unknown. RecT (269aa), like λ Beta, is an SSAP [12]. Phage λ also encodes an inhibitor of the *E. coli* RecBCD nuclease, the Gam protein [13], which allows preservation of linear dsDNA in the bacterial cell. The Rac prophage does not contain a *gam* gene, although it has been suggested that an equivalent inhibitory function is encoded within the large *recE* gene [14]. λ *gam* has been included in some RecET expression constructs and increases RecET-mediated recombination frequencies in some assays [14,15]. While RecE and RecT play analogous roles to λ Exo and Beta [1,12], and RecET can replace Exo and Beta for λ growth and recombination [12,16], there is little sequence identity between the two systems other than a few key amino acids [17,18]. Further, each protein only functions with its genetic companion [4], i.e., RecE only functions with RecT, and not with Redβ. Thus, while both systems can be used in *E. coli* for recombineering, they may differ in their mechanistic details.

Homology-dependent *in vivo* recombination enables the efficient assembly of multiple linear dsDNA fragments into a single DNA molecule, typically a plasmid. [15]. The linear DNAs containing the required homologies are introduced into recombination-proficient cells by transformation [19]. *In vivo* cloning has been adapted for bioprospecting [15] and complex plasmid constructions. This method allows simultaneous incorporation of several genetic elements, such as promoters, genes, and gene tags. The technique avoids the difficulty of sequentially cloning these individual elements and can provide a simple and inexpensive *in vivo* alternative to Gibson Assembly [20].

Both the Red and RecET recombination systems can perform this homology-dependent reaction, but the RecET system exhibits superior recombination efficiency compared to the Red system [15]. Linear DNA assembly can also be achieved in JC8679, a strain that expresses RecET [21], or in DH5α [22–24], a commercially available strain commonly used for plasmid propagation and cloning. Nozaki and Niki [23] have studied and optimized linear DNA assembly in DH5α and other *E. coli* strains, expanding the applicability of *in vivo* recombination for molecular cloning.

In this paper, we report characterization and optimization of various parameters of the *in vivo* linear DNA assembly reaction mediated by RecET and λ Red. With

both systems, we find that deletion of the *E. coli xonA* gene, encoding Exonuclease I (ExoI), a 3'→5' ssDNA exonuclease [25], improves the efficiency of this DNA assembly reaction. The RecET system allows robust *in vivo* assembly of at least six DNA fragments in a single reaction, with very few errors. This discovery is a substantial improvement on *in vivo* linear DNA assembly as compared to other published systems [15,21–24,26–30], exhibiting similar efficiency and accuracy as the most widely used *in vitro* method for assembling multiple DNA fragments, Gibson Assembly [20].

## Materials and methods

### Strains, plasmids, and growth conditions

Bacterial strains are derivatives of *E. coli* K-12 (Table 1). LB media, both liquid and solid agar plates, were used for growth. Ampicillin was used at 100 µg/ml. To test the antibiotic resistance of recombinant plasmids, Amp$^R$ colonies were patched to LB plates containing either kanamycin at 30–50 µg/ml or chloramphenicol at 10 µg/ml. To maintain pSIM26, tetracycline was used at 25 µg/ml in liquid LB medium. To screen for blue colonies indicative of correct *lacZ* assembly and β-galactosidase activity, X-gal was added to LB solid agar plates at a final concentration of 20 µg/ml.

### Generation of linear fragments

Linear DNA fragments were generated by PCR using Platinum Taq HiFi (Invitrogen) or Q5 High-Fidelity 2x Master mix (NEB) and purified using a QIAquick PCR Purification kit (Qiagen) or NucleoSpin Gel and PCR cleanup kit (Macherey-Nagel). DNA oligonucleotides (oligos) and gBlocks were obtained from IDT (Integrated DNA Technologies). Some oligonucleotides were obtained from Eurofins (Eurofins Genomics). Sequences of DNA oligos are in S2 Table. Fragments contained 50 bp of terminal homology unless otherwise noted. Fragments were generated from linear templates as described in Supplementary information, except for *lacZ* fragments, which were generated by colony PCR [19]. Purified linear DNAs were introduced individually into the relevant host by electroporation, and ampicillin selection was applied to ensure the absence of intact plasmid

**Table 1. *Escherichia coli* K-12 strains and plasmids.**

| Strain name | Relevant genotype | Source |
|---|---|---|
| HME6 | W3110 Δ*lacU169 galK*$_{TYR145UAG}$ λcI857 Δ*(cro-bioA)* | [3] |
| JS663 | W3110 Δ*lacU169 galK490* [λ *cI857* (*int-cIII<>cat exo bet Δgam*) Δ(*cro-bioA*)] | this study |
| JS677 | JS663[pSIM26] Tet$^R$ (Gam producing plasmid) | this study |
| JS679 | *W3110 ΔlacU169 galK*$_{TYR145UAG}$ [λ *cI857 recE recT* Δ(*cro-bioA*)] [pSIM26] Tet$^R$ | this study |
| JS726 | JS679 *recA<>spec* | this study |
| JS727 | JS677 *recA<>spec* | this study |
| LT1795 | W3110 Δ*lacU169 galK*$_{TYR145UAG}$ λcI857Δ (*cro-bioA*) [λ (*int-cIII*)<>*gam recE recT*] | [31] |
| NC540 | W3110 Δ*lacU169 galK*$_{TYR145UAG}$ Δ*xonA* λcI857Δ(*cro-bioA*) | this study |
| NC553 | W3110 Δ*lacU169 galK*$_{TYR145UAG}$ Δ*xonA* λcI857 Δ(*cro-bioA*) (*int-cIII*)<>*gam recE recT* | this study |
| SIMD63 | W3110 Δ*lacU169 galK*$_{TYR145UAG}$ λcI857 Δ (*cro-bioA*) [λ (*int-cIII*)<>*recE recT*] | [19] |
| **Plasmid name** | **Relevant genotype** | **Source** |
| pLT59 | pUC plasmid *bla$^+$kan$^+$* | [32] |
| pJS103 | pUC plasmid *bla$^+$lacZ kan$^+$* | this study |
| pSIM26 | pSC101 *gam$^+$tet$^+$* | This study |

template contamination. In fragment assembly experiments, 100 ng of each linear DNA was used unless noted. To prevent arcing during electroporation, in six-fragment assembly assays, equal amounts of individual fragments at 100 ng/μL were mixed, keeping the overall DNA concentration at 100 ng/μL but each fragment at 16.7 ng/μL. Cells were electroporated with 2 μl of this mixture; thus, the final concentration of each fragment was 33 ng/μL for six-fragment assembly experiments.

### Expression of recombination functions and electroporation

Cells were induced for the recombination functions and prepared for electroporation as previously described [19]. Following introduction of the DNA by electroporation, cells were outgrown in 1 ml LB for 2 hrs at 32 °C, diluted and plated on solid LB medium to score total viable cells, and on LB + Amp to score Amp$^R$ plasmid recombinants. Petri plates were incubated at 32 °C. In some experiments, Amp$^R$ colonies were patched to either LB + Kan or LB + Cm to screen for resistance as an indication of assembly accuracy. The frequency of fragment joining was calculated as (Amp$^R$/Viable cells) (1x10$^8$); normalized to 1x10$^8$ cells, approximating the number of cells that survive electroporation under our experimental conditions.

### Conditions for comparison of NEBuilder® HiFi DNA Assembly and RecET Δ*xonA* assembly

We used 30 bp homology arms, within the 20–30 bp range commonly used in NEBuilder® HiFi DNA Assembly, as this length yielded robust assembly efficiency in our system. NEBuilder® HiFi DNA Assembly ideally uses ~0.05 pmol of each fragment (0.03–0.2 pmols for the mix in equal molar ratio). For three fragments, we used 100 ng of each fragment, which works out to: fragment 1 (3395 bp – 0.05 pmol), fragment 2 (1131 bp – 0.14 pmol), and fragment 3 (1688 bp – 0.10 pmol). The identical mix was used for assembly by our system using protocols described herein, or by NEBuilder® HiFi DNA Assembly (NEB catalog #E2621) following the manufacturer's instructions. For three fragments, the NEBuilder® HiFi DNA Assembly was performed at 50 °C for 15 minutes, and 2 μl of the assembly mix was transformed into NEB5α high-efficiency cells (NEB catalog #C2987H) following the manufacturer's protocol. For six fragments, NEBuilder® HiFi DNA Assembly recommends ~0.05 pmol of each fragment (0.2–0.5 pmols for the mix in equal molar ratio) but suggests up to 10x higher for short fragments. We maintained that concentration, applying the additional criterion that each DNA fragment was used at no less than 30 ng. Thus, for short fragments, the concentration was higher as follows: fragment 2 (552 bp – 0.09 pmol), fragment 4 (100 bp – 0.5 pmol), fragment 4 (400 bp – 0.12 pmol). When used, 5 pmol of the 100-nucleotide oligo was supplied. All other procedures for six-fragment assembly were the same as for three fragments, except for the NEBuilder® HiFi DNA Assembly reaction incubation time, which was 1 hour at 50 °C following the manufacturer's recommendation.

### Analysis of pBR-*lacZ* plasmids from transformants

In selection experiments, white (Lac$^-$) Amp$^R$ colonies were isolated on LB + Amp + X-gal. Single colonies were used to inoculate 5 ml overnight cultures in LB broth containing ampicillin, and plasmid DNA was subsequently purified using a Qiagen Miniprep Kit. Plasmids were digested with *Pst*I, which has a single restriction site in the *bla* gene, and DNA was analyzed on agarose gels to verify that the recovered plasmids were of the expected size. Some plasmids were further analyzed by sequencing the *lacZ* gene (S3 Fig).

## Results and discussion

### *In vivo* assembly of two linear dsDNAs

We first confirmed published observations [15] using the Red and RecET systems. Initially, strains expressing either the λ Red or RecET functions were compared for their ability to assemble plasmids *in vivo* from two linear DNAs introduced into cells by electroporation. Recombination functions were expressed from the λ $P_L$ promoter in single copy on the bacterial

                                  

chromosome, under control of the temperature-sensitive CI857 repressor (Fig 1A), and standard recombineering tech-niques were used [19]. One strain (HME6) contains the phage λ Red system, and another (LT1795) contains the RecET system; both strains express λ *gam*. λ Red (JS663) and RecET (SIMD63) strains lacking *gam* were also tested.

In these experiments, two dsDNA fragments were used to generate an intact plasmid (pLT59). Each DNA fragment contained a partial origin of DNA replication (*ori*) and a partial kanamycin resistance (*kan*) gene (Fig 1B); neither fragment individually can generate a replicating plasmid. Intact plasmids were selected by resistance to ampicillin (Amp^R: encoded by the *bla* gene, found on one of the fragments). Only precise joining of the two DNAs will generate a functional plasmid *ori* and a Kan^R gene. The results are shown in Fig 1C.

As previously reported [15], the RecET system is up to 1000-fold more efficient at *in vivo* linear DNA assembly than is the λ Red system (Fig 1C, Red vs. RecET). When Amp^R colonies were patched to LB agar containing kanamycin, the frequency of Kan^R was > 95% for all strains tested, indicating accurate joining of the junction within the *kan* gene by either recombination system. Expression of Gam stimulated λ Red-dependent recombination and plasmid assembly ~10-fold,

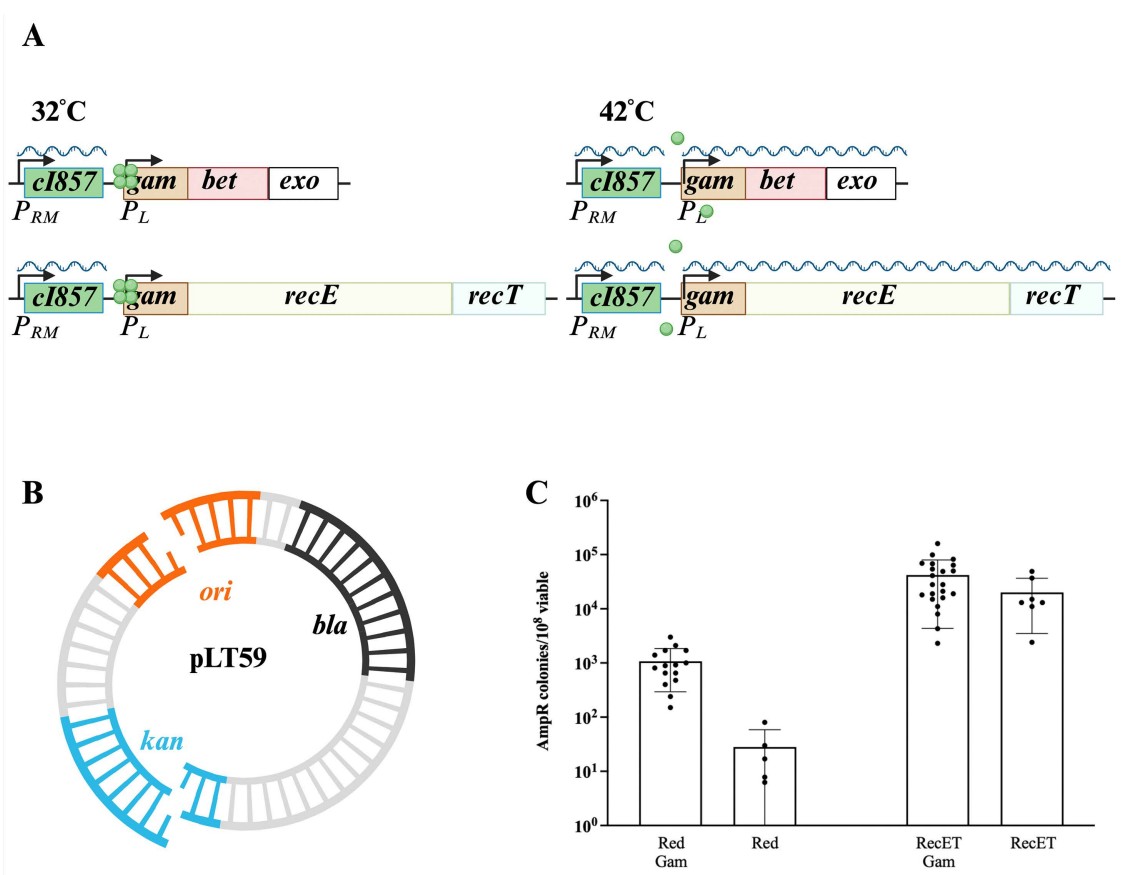

**Fig 1. Assembly of two linear DNAs into intact plasmids by λ Red and RecET systems. A.** Expression of λ Red or RecET from the $P_L$ operon of a defective λ prophage. At 32 °C, the CI857 repressor protein (green circle) is functional and the *red* or *recET* genes are not transcribed. At 42 °C, the CI857 repressor unfolds and no longer binds to the operators allowing transcription of the recombination genes from the $P_L$ promoter. **B.** Two-fragment linear DNA assembly. 50 bp of terminal homology is present in each fragment within the *ori* (origin of DNA replication) or the *kan* gene, indicated as the single-strand bases. *bla* encodes β-lactamase which when expressed results in ampicillin resistance (Amp^R) **C.** Recombinant frequency is expressed as the number of Amp^R colonies/10^8 total colonies for the indicated recombineering systems. Error bars indicate standard deviation (s.d.). Results from independent experiments are indicated by the black dots. Assembly was dependent on supplying both DNA fragments and on expression of the λ Red or RecET functions.

as expected from previous recombination experiments [33]. Interestingly, the presence or absence of Gam did not significantly alter the frequency of linear DNA assembly by RecET, in agreement with a proposal [14] that the larger RecE protein may contain a Gam-like activity. However, from an abundance of caution, in all other RecET experiments described here, we used the Gam-expressing strains to ensure protection of linear DNA from the RecBCD nuclease, which ordinarily would digest the linear DNA substrates.

## Effect of increasing terminal homology length on linear DNA assembly

We observed that extending the length of terminal homology progressively increased the frequency of λ Red-mediated recombination such that with 200 bp of homology, Red-mediated recombination is only about 10-fold lower than that achieved with RecET, and with more than 330 bp of homology, the difference between the two systems is only 5-fold (Fig 2). We also found that the increased recombination frequency observed with longer homologies is not dependent on *E. coli* recombination systems, since mutation of *recA* did not affect the frequency. The enhancement of Red recombination for linear DNA assembly seen with longer homologies is consistent with our previous results [31], in which we tested *in vivo* Red- and RecET-mediated intramolecular circularization of a linear dimer plasmid to form a circular monomeric product. This reaction provided extremely long terminal homologies, with a final circular product of ~4.4 kb and an initial linear substrate of twice that length. This reaction was extremely efficient for both Red and RecET, and the two systems gave similar frequencies for this recombination. The requirement of longer homologies for the Red system and the observed differences in plasmid allele inheritance [31] suggest that the two recombination systems process linear substrates differently.

## Mutation of Exonuclease I function improves recombinant yield for linear DNA assembly

Processing of blunt linear dsDNA ends by either λ Exo or RecE leaves 3' ssDNA overhangs. If not bound and protected by Beta or RecT, these single strands could be substrates for bacterial 3'→5' exonucleases. Thus, degradation by a host 3'→5' exonuclease could lower recombination efficiency by removing the DNA substrate. Since ExoI is a major *E. coli* 3'→5' ssDNA exonuclease [25], we asked whether removing ExoI function by deletion of the *xonA* gene improves the efficiency of linear DNA assembly. We tested the recombination proficiency of strains expressing either λ Red (NC540) or RecET (NC553) deleted for the *xonA* gene (ΔxonA). As shown in Fig 3, removal of ExoI activity enhanced recovery of recombinant plasmids nearly ~65-fold for the λ Red system and >70-fold for RecET. Amp$^R$ colonies from the strains deleted for *xonA* were patched to L+Kan; >97% of the isolates were Kan$^R$, thus the loss of ExoI function did not affect the fidelity of DNA fragment assembly.

We also tested the ability of the RecET strain to assemble three DNA fragments to make the same intact plasmid (pLT59, S1 Fig). As before, the DNA junctions bisected *ori* and the *kan* gene. In this case, the *bla* gene encoding ampicillin resistance was also cleaved so that accurate assembly was necessary not only for plasmid replication, but also for resistance to ampicillin (kanamycin resistance was not scored in these experiments). The three fragments were efficiently assembled by the RecET system, and elimination of the ExoI function increased the assembly frequency by ~20-fold (Fig 3).

## Two fragment assembly: Dependence on substrate DNA concentration

Since the RecET system is more efficient than the λ Red system for linear DNA assembly, we examined other aspects of RecET-mediated recombination. To ask whether recombination efficiency changes with substrate DNA concentration, a range of linear DNA concentrations was tested in both the *xonA*$^+$ and ΔxonA RecET expressing strains (Fig 4). The ΔxonA strain showed robust levels of fragment joining even at the lowest DNA concentration tested, 10 ng per fragment.

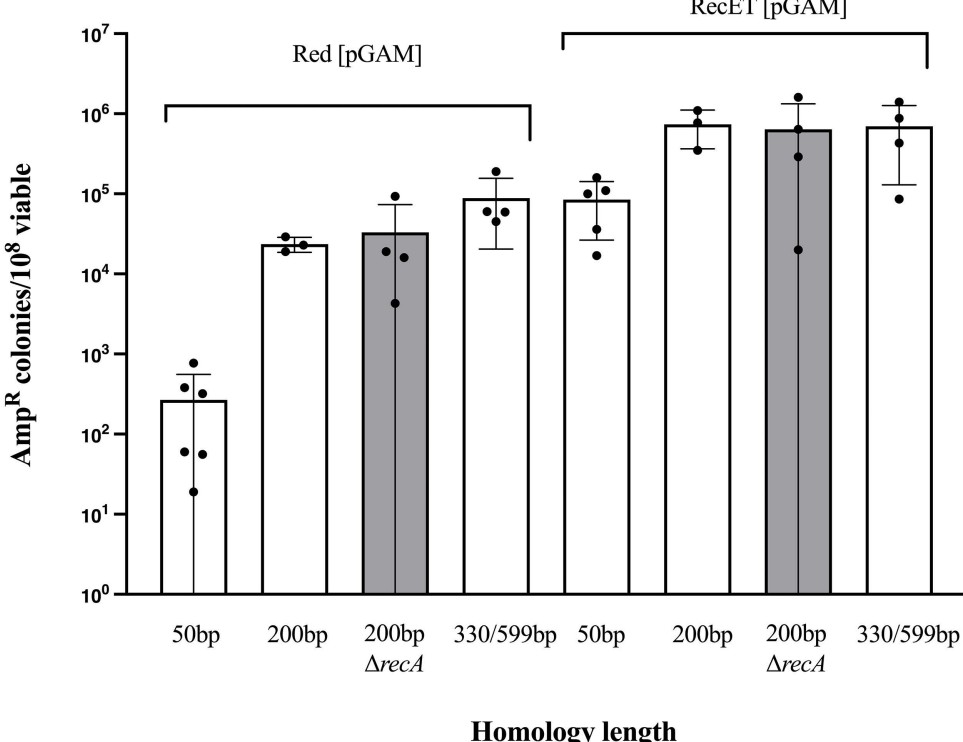

**Fig 2. Assembly of two linear DNAs with various homology lengths into intact plasmids by the λ Red and RecET systems.** Recombinant frequency is expressed as the number of Amp$^R$ colonies/$10^8$ total colonies for the indicated recombineering systems. In all cases, λ Gam protein was supplied from the low-copy tetracycline resistant plasmid pSIM26. Homology lengths are indicated on the x-axis, in one case they were 330 bp in the plasmid origin and 599 bp in the *kan* gene. In the grey bars, the cells are also mutant for *recA*. Results from independent experiments are indicated by the black circles. Assembly of pLT59 was dependent on supplying both fragments and on expression of the λ Red or RecET functions. Error bars indicate s.d.

## Two fragment assembly: recombination dependence on homology length

We asked whether shorter homology lengths are efficient for fragment joining, using a similar design to assemble two linear DNAs, as shown in Fig 1B. With PCR, we amplified pairs of linear dsDNAs with terminal homologies of various lengths, and tested these fragments in the RecET *xonA$^+$* vs. the Δ*xonA* strain. As shown in Fig 5, 30 bp of terminal homology provides near maximal recombination capability, similar to results seen previously in DH5α [23,24,26]. Our RecET results are similar to those of Fu *et al.* [15] (see their Supplemental Fig 1), where they observed an increase in recombinant frequency as homology length increased from 50 bp to 120 bp. However, homologies longer than 30–50 bp are not required for efficient fragment assembly in the RecET Δ*xonA* strain and thus PCR products can be generated using DNA oligos as short as 50 nt in total length (30 nt homology plus 20 nt for priming).

## *In vivo* linear dsDNA assembly of plasmids from six independent linear dsDNA fragments

To test assembly of six linear dsDNA fragments, we designed a plasmid, pBR-*lacZ*, with the *E. coli lacZ* open reading frame replacing the tetracycline resistance gene, *tet*, of pBR322 (Fig 6A). The three fragments comprising the *lacZ* gene were either made by PCR or synthesized as gBlocks. The plasmid backbone was contained in three other PCR fragments. After the recombination reaction, cells were plated on LB + Amp + X-gal to screen for blue colonies indicating β-galactosidase activity and thus accurate assembly of the *lacZ* gene. When six DNA fragments were used to assemble

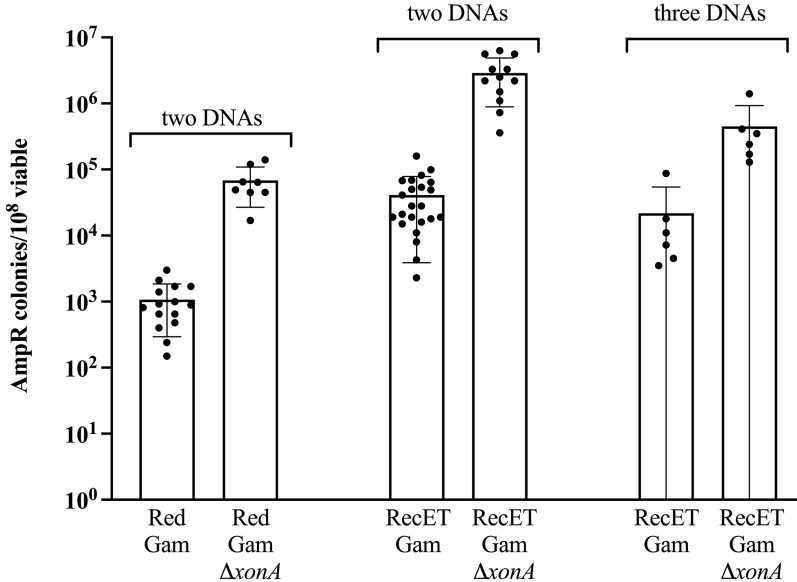

**Fig 3. Removal of Exonuclease I activity improves recombinant yield.** The recombinant yield is expressed as the number of Amp$^R$ colonies/10$^8$ total colonies for the Red and RecET systems, both with Gam expressed. Results from independent experiments are indicated by the black dots. Error bars indicate s.d. All homology lengths are 50 bp.

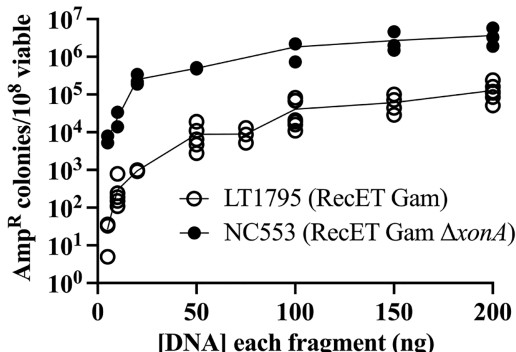

**Fig 4. Dependence of two-fragment linear DNA assembly on DNA concentration.** The data show number of Amp$^R$ recombinants/10$^8$ viable cells over a range of DNA concentrations. The concentration of each linear dsDNA is indicated. Results from independent experiments are indicated by the circles. Closed circles (●) indicate RecET Δ*xonA* (NC553); open circles (○) indicate RecET *xonA*$^+$(LT1795).

pBR-*lacZ* in the Δ*xonA* strain expressing RecET, the recombinant frequency was nearly 4.0x10$^4$ per 10$^8$ viable cells (Fig 6B), only ~10-fold lower than the frequency of the three fragment reaction in the Δ*xonA* background (compare to Fig 3). Reducing the DNA concentration of each fragment from 100 ng to 33 ng (Materials and methods) may be responsible for much of this reduction in frequency (Fig 4). When all fragments were generated by PCR, ~90% of the Amp$^R$ colonies obtained from the Δ*xonA* strain were blue on LB + Amp + X-gal indicator agar, demonstrating β-galactosidase activity and thus correct assembly of the *lacZ*$^+$gene. When the three fragments encoding *lacZ* were synthesized as gBlocks (IDT), 84% of the colonies were blue on LB + Amp + X-gal indicator agar. We isolated a total of 113 plasmids from independent white colonies from either the PCR or PCR/gBlock reactions performed in the Δ*xonA* background. These plasmids were

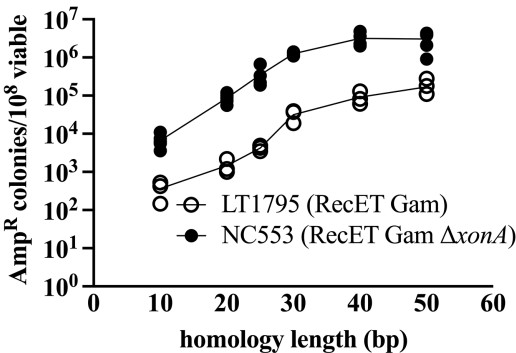

**Fig 5. Dependence of two-fragment linear DNA assembly on terminal homology length.** The data show the number of Amp^R recombinant plasmids/$10^8$ viable cells obtained over a range of homology lengths. Results from independent experiments are indicated by the circles. Closed circles (●) indicate RecET ΔxonA (NC553); open circles (○) indicate RecET xonA⁺(LT1795).

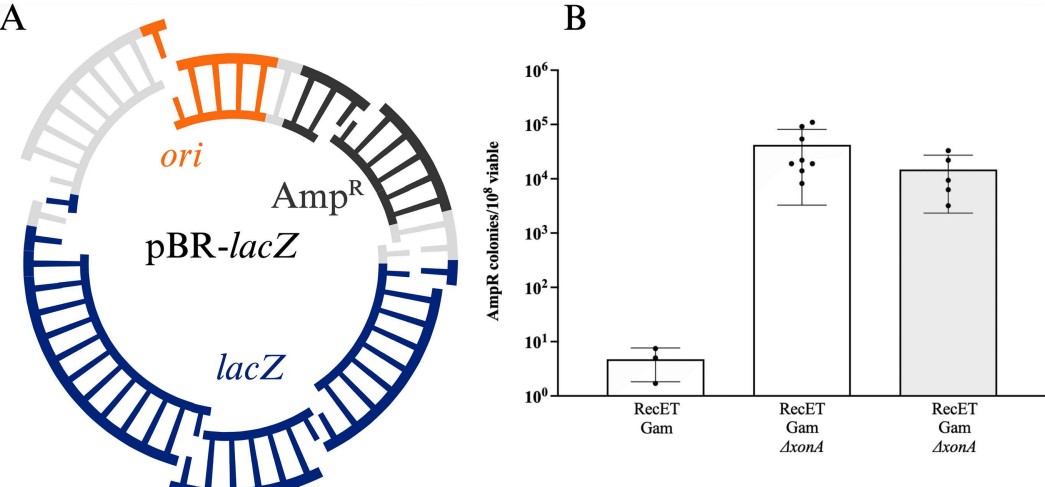

**Fig 6. Assembly of six linear DNA fragments by RecET. A.** Diagram of six-way plasmid assembly for pBR-*lacZ*. Terminal homologies present in each fragment within the *ori*, *bla*, and *lacZ* genes are indicated by the single-strand bases. One single-strand base indicates a homology of 50-60 bp whereas two single-strand bases indicate a homology of ~100 bp. Similar results were seen when all homologies were 50 bp (S2 Fig). **B.** Recombinant frequency obtained for six-way plasmid assembly under *xonA⁺* (LT1795) and Δ*xonA* (NC553) conditions is expressed as the number of Amp^R recombinant plasmids/$10^8$ viable cells. All experiments used linear dsDNA generated with PCR except for the results shown with the gray bar, where the three linear DNAs comprising the plasmid backbone were made with PCR and the three comprising the *lacZ* gene were synthesized as gBlocks. Results from independent experiments are indicated by the black dots. Error bars indicate s.d.

analyzed by restriction analysis (Materials and Methods); in all but one case, the six fragments had assembled correctly. Forty of the 113 plasmids isolated from white colonies were further examined by sequencing the *lacZ* gene and flanking region; this analysis revealed that most mistakes arise during PCR or gBlock synthesis rather than occurring at the recombinant junction (S3 Fig). In contrast, for LT1795, the *xonA⁺* strain, increasing the number of linear DNA fragments from three to six caused a ~ 340-fold reduction in frequency, with only ~25% of the colonies blue on X-gal indicator. Restriction analysis of four white colonies from this strain revealed incorrect assembly of the plasmid. Thus, the Δ*xonA* mutation is essential for accurate assembly of six or more fragments by the RecET system.

Our results (Fig 6B) show that elimination of the host ExoI nuclease function by deletion of the *xonA* gene substantially increases the efficiency of linear DNA assembly, allowing at least six fragments to be assembled by the RecET system into a functional plasmid (see S2 Fig for additional data). Using the same rigorous test of assembling six linear dsDNA fragments, we asked whether the removal of other host ssDNA exonucleases impacted plasmid assembly. Removal of either the RecJ or ExoX functions did not increase recombination frequencies in the way that removing ExoI did (**S1 Text. Supporting results and discussion.**).

### How does RecET-mediated Δ*xonA* fragment assembly compare to other *in vivo* systems?

Over the years, multiple methods have been developed for *in vivo* plasmid assembly. These methods have allowed assembly of complex plasmids to be designed and constructed with precision and accuracy. Regulatory elements, gene tags, fusions, and antibiotic resistance genes can all be introduced onto a plasmid backbone in a single round of *in vivo* cloning. However, as the number of fragments to be assembled increases, cloning efficiency can rapidly decrease [22,23,26–28]. Previous methods of *in vivo* cloning fall into two categories: either using a phage recombination system or an endogenous recombination activity present in some laboratory bacterial strains, such as DH5α.

Fragment assembly in DH5α has been developed by several groups [22–24,26]. The ability to assemble linear dsD-NAs has also been seen in other common laboratory bacterial strains [34]. This method requires a minimum of 10 bp of homology, with 30 bp showing optimal results [26]. Surprisingly, electrocompetent cells do not work well for this system; rather, chemically competent cells are needed [22–24,26]. Comparing assembly efficiencies can be difficult because of the non-uniform presentation of results, but it appears that the best published [22] frequency of assembling 2 fragments reported was ~$8 \times 10^4$ recombinants for 100 ng of vector. In contrast, our method produced ~$6 \times 10^5$ assembled plasmids with the same amount of DNA. Once more fragments are assembled, i.e., 6, the difference in frequency is more pronounced. Whereas we generated ~$2 \times 10^4$ plasmid clones from 100 ng of vector, only 4–10 correct clones were recovered from 6 fragment assembly in DH5α [22], suggesting that in this system, the reaction is approaching the limits of recoverability.

The RecET recombination system, active in JC8679 cells, has been shown to assemble two fragments *in vivo* at a frequency of as high as ~$4 \times 10^4$ for 100 ng of vector [21]. The data of Fu *et al.* [15] and Baker *et al.* [28] also show robust results for two and four fragments with the RecET system, but fall off sharply when assembly of five different fragments is attempted. This is consistent with our observations in a RecET *xonA*+ strain (Figs 1C, 3, 6B).

Recently, combinations of two different exonucleases and three different ligases were expressed both in *E. coli* and in other bacteria in order to assemble dsDNA fragments. In the best case, using T5 exonuclease and T4 DNA ligase, this phage enzyme-assisted *in vivo* DNA assembly (PEDA) produced up to $3.5 \times 10^4$ plasmid clones per µg vector DNA when two fragments were joined using 40 bp homologies [30]. This is approximately 1000-fold less than the plasmid clones that we generate if we used 1 µg vector DNA. Based on our results, we hypothesize that deleting the *xonA* gene in their system would also enhance the recovery of recombinant plasmids.

### How does RecET-mediated Δ*xonA* fragment assembly compare to other fragment assembly methods?

Besides traditional cloning with restriction enzymes and ligase, there are many methods for dsDNA assembly of multiple fragments that include *in vitro* steps. One of the most widely utilized and best methods for linear DNA assembly is Gibson Assembly [20]. This technique involves processing linear dsDNA fragments with overlapping terminal homologies (20–40 bp) *in vitro* using purified proteins, including exonuclease, polymerase, and ligase activities, resulting in the fusion of these fragments into a single continuous strand. Subsequently, in a second step, the mixture is transformed into *E. coli* to isolate plasmid candidates. In Fig 7, we present a direct comparison of the plasmid generation frequency from fragment assembly using NEBuilder® HiFi DNA Assembly (a commercially available Gibson Assembly variant kit) and RecET-mediated Δ*xonA* fragment assembly, employing identical fragment mixes. The fragments possess 30 bp homology at their ends, and the assembly of either three or six fragments is evaluated. The plasmid, pJS103, encodes

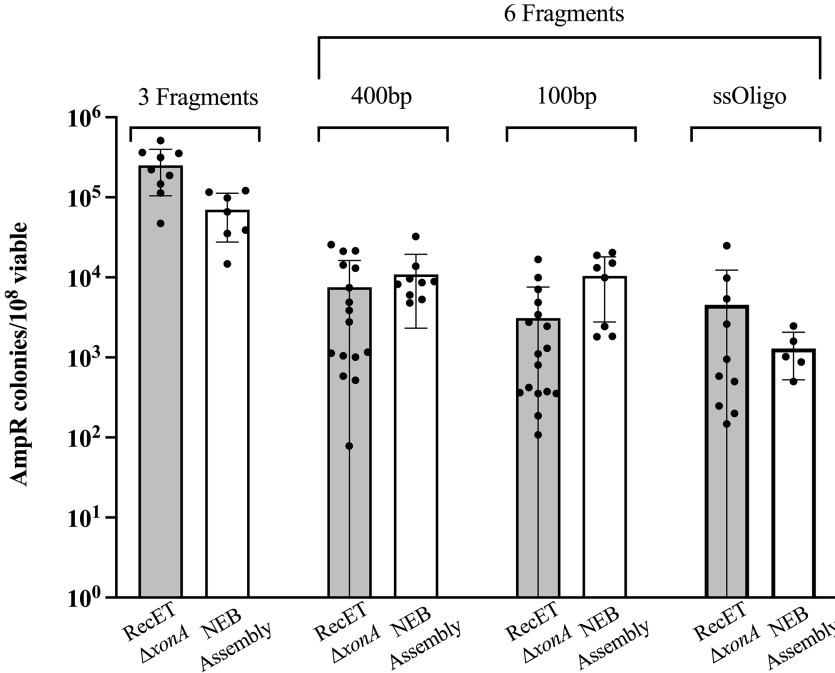

**Fig 7. Assembly of linear DNA fragments by RecET vs NEB Asembly.** Fragments were used to assemble pJS103 (S4 Fig). Either 3 fragments (first two bars) or six fragments were used to assemble the plasmid. "400 bp", "100 bp", and "ssOligo" refers to the size of the smallest fragment of the six. All fragments had 30 bp of homology to their partner fragment. Shaded bars indicate plasmids formed from RecET-mediated ΔxonA assembly, and white bars indicate plasmids formed by NEBuilder® HiFi DNA Assembly (shortened to "NEB assembly" for simplicity) following the manufacturers protocols including transformation to commercial NEB5α cells (Materials and Methods). The recombinant yield is expressed as the number of AmpR colonies/$10^8$ total colonies. Results from independent experiments are indicated by the black dots. The error bars indicate s.d.

the *E. coli lacZ* gene, along with ampicillin- and kanamycin-resistance genes (see Materials and Methods and S4 Fig). Following the assembly reaction, cells are plated on LB+Amp+X-gal to screen for blue colonies, indicative of β-galactosidase activity and hence accurate assembly of the *lacZ* gene. Subsequently, these colonies were patched onto LB+Kan to confirm the accurate assembly of the kanamycin-resistance gene (Table 2). Efforts were made to maintain optimal conditions for both reactions (Materials and Methods). Our data reveal that both methods yield similar results for frequencies and accuracy. Upon closer inspection, it is apparent that NEBuilder® HiFi DNA Assembly demonstrates a slight advantage with very small fragments (100 bp), suggesting that RecET-mediated assembly may favor longer fragments. However, our results also illustrate that a synthetic 100 nucleotide oligo can serve as a substrate for RecET-mediated ΔxonA assembly, consistent with previous results for RecET [28]. Both methods are highly accurate (Table 2), yet in the rare cases when a white colony is found, it is more likely that the plasmid is kanamycin-sensitive as well, indicating incorrect plasmid assembly.

Other *in vitro* methods, including SLiCE [35] and ZeBrα [27], utilize Red recombination proteins *in vitro,* followed by a transformation step into competent *E. coli.* The levels of assembly, however, were 50- to 100-fold lower than our RecET-mediated ΔxonA fragment assembly with a given number of fragments. Another method, SLIC, uses an exonuclease (T4 polymerase) to process the linear fragments *in vitro,* leading to ssDNA that can anneal and be transformed to isolate recombinants [36]. Again, the RecET-mediated ΔxonA fragment assembly reported in this work is up to 100-fold more efficient than SLIC. A variation of SLIC, ExoCET cloning, also requires *in vitro* treatment of the DNA with T4 polymerase. It was developed to increase the frequency of "direct DNA cloning", i.e., retrieval of a specific region of DNA from a complex genome [37]. This method allows retrieval from a genomic mix at a frequency of up to $1.6 \times 10^4$ (normalized to 100 ng vector). Song et al. [29]

**Table 2. Accuracy of fragment assembly methods.**

| Assembly Method | # Fragments (shortest fragment) | % Blue[1] | Among the Blue % KanR[2]Blue | Among the White % KanR[3] |
|---|---|---|---|---|
| RecET Δ*xonA* | 3 (1131) | 99.2 | 99.7 | 60 |
| NEB assembly | 3 (1131) | 96.7 | 100 | 18 |
| RecET Δ*xonA* | 6 (400) | 95.4 | 98.8 | 82.9 |
| NEB assembly | 6 (400) | 99.4 | 98.7 | 60 |
| RecET Δ*xonA* | 6 (100) | 96.0 | 99.0 | 74.4 |
| NEB assembly | 6 (100) | 96.9 | 98.1 | 68.5 |
| RecET Δ*xonA* | 6 (ssOligo) | 94.7 | 98.6 | 59.0 |
| NEB assembly | 6 (ssOligo) | 96.1 | 98.6 | 55.8 |

1 Average % blue colonies from all experiments in Fig 7.

2 Average % kanamycin-resistant colonies among those that were blue from all experiments in Fig 7.

3 Average % kanamycin-resistant colonies among those that were white from all experiments in Fig 7.

showed that, for the direct assembly of multiple fragments (not a complex mix), ExoCET was more effective than Gibson Assembly in producing clones containing all intended fragments. Seven fragments were assembled at a frequency of $1.5 \times 10^2$ (normalized to 100 ng vector); and even 13 fragments could be assembled at a frequency of $7.3 \times 10^1$. Our experiments raise the interesting question as to whether a *xonA* mutation would enhance the efficiency of the ExoCET reaction.

## Conclusions and perspectives

Our data demonstrates that use of a strain expressing RecET and deleted for *xonA* strain gives high levels of accurate recombination using nanogram quantities of substrate DNA and is likely to be the method of choice for *in vivo* assembly of more than two fragments simultaneously. The technique does not require special vectors, extract preparation, or *in vitro* reactions before transformation into competent cells. It compares favorably with NEBuilder® HiFi DNA Assembly both in frequency and accuracy. We have used the RecET Δ*xonA* strain to make a plasmid pool for a two-hybrid analysis by recombining a linearized vector with a PCR-generated DNA [38]. The high efficiency of our system generates many recombinants for analysis. These recombinant clones can be identified phenotypically, with a colony PCR screen, or by plasmid DNA isolation and restriction analysis (Supplementary Information). Like others [15], we have observed that for RecET the recombining homology need not be exactly at the end of the linear DNA and that terminal non-homologies (from a few bases to >5 kb) will be removed during the recombination process, unlike ExoCET recombination [29] or NEBuilder® HiFi DNA Assembly, which will remove up to 10 bp (NEB website). Although we have not tested assembling more than six DNA fragments into a single plasmid, we predict that it will be possible in the RecET Δ*xonA* strain. We propose that the Δ*xonA* mutation allows the 3' ssDNA ends produced by RecE to persist in the cell, allowing enhanced RecT-dependent annealing of complementary sequences to form accurate recombinant products with high efficiency.

## Supporting information

**S1 Text. Supporting results and discussion.**
(PDF)

**S1 Fig. DNA fragments used for the three-way assembly of pLT59.** Terminal homologies present in each fragment within the *ori*, *bla*, and *kan* genes are indicated by the single-stranded bases. One single-strand base indicates a homology of 50–60 bases. The data are shown in Fig 3 of the main paper.
(PDF)

**S2 Fig. Additional data for *in vivo* linear assembly from six linear dsDNA fragments. A.** Six PCR fragments were used to assemble the circular plasmid pLT61. Of the six junctions, one is within the plasmid *ori*, a second is within the *bla* gene, and two more are within the chloramphenicol resistance gene, *cat*. **B.** Elimination of the host ExoI function (i.e., Δ*xonA*) increased the frequency of plasmid assembly ~1000 fold. When Amp$^R$ colonies from the RecET Δ*xonA* recombination were scored for Cm$^R$, 194/200 colonies were Cm$^R$, indicating accurate joining of the linear DNA segments. We anticipate that most of the Cm$^S$ plasmids arose from PCR mistakes or primer synthesis mistakes, as found for the *lacZ* gene in other six-way assembly experiments (see Fig 6, main paper).
(PDF)

**S3 Fig. Sequence analysis of white colonies from the pBR-*lacZ* assembly reactions.** The black bar with cyan segments represents a portion of the pBR-*lacZ* plasmid. The homology overlaps between fragments used for assembly are shown in cyan. The 3075 bp *lacZ* open reading frame *(orf)* is indicated by the grey arrow. The region from the promoter to the translational stop was sequenced; locations of observed mutations are indicated with vertical lines and a colored circle with a number, indicating the isolate number. If multiple mutations were found in an isolate, they are indicated with the same color and number. We did not find increased numbers of mutations in the overlapping homology region where the single-strand annealing occurs **A.** PCR products were used for assembly. A total of 20 isolates were sequenced and among these, 68 point mutations and 6 insertion or deletions were found. 9 candidates had mutations in the overlap homologies. **B.** gBlocks were used for assembly. A total of 20 isolates were sequenced and among these, 17 point mutations and 14 insertion or deletions were found. Only 3 isolates had any mutations in the overlap homologies. The 5 candidates not shown were isolates that had apparent synthesis errors containing small repeats.
(PDF)

**S4 Fig. DNA fragments used for the six-way assembly of pJS103.** Terminal homologies present in each fragment within the *ori, kan, lacZ,* and *bla* genes are indicated by the single-strand bases. One single-strand base indicates a homology of 50 bases. Fragment #4 (indicated with a "4") is always the smallest fragment and, in different experiments, was 100 bp, 400 bp, or even a 100-base oligo as shown in the data in Figure 7 of the main paper. The length of fragment 5 (within *lacZ*) varies according to the length of fragment 4; when fragment 4 is longer, fragment 5 is shorter, etc.
(PDF)

**S1 Table. *Escherichia coli* K-12 strains and plasmids for supporting information.**
(PDF)

**S2 Table. Oligos used.**
(XLSX)

## Acknowledgments

We thank Carolyn Court for careful reading of the manuscript and insightful comments. We also thank M. Spencer, N. Shrader, T. Hartley, and K. Pike from the CRTP Genomics Laboratory at the Frederick National Lab for Sanger sequencing. All images in figures were created in their entirety by the authors as follows: Figs 1A, 1B, 6A, S1, S2A and S4 were created in https://BioRender.com; Figs 1C, 2, 3, 4, 5, 6B, 7, S2B were made with Graphpad Prism; and S3 Fig was generated using Snapgene.

## Author contributions

**Conceptualization:** James Sawitzke, Nina Costantino, Lynn C. Thomason, Donald L. Court.

**Funding acquisition:** James Sawitzke, Donald L. Court.

**Investigation:** James Sawitzke, Nina Costantino, Adriana Castillo Caballero, Ellen Hutchinson, Alessandro Barenghi.

**Methodology:** James Sawitzke, Nina Costantino, Lynn C. Thomason, Donald L. Court.

**Project administration:** James Sawitzke, Donald L. Court.

**Supervision:** James Sawitzke, Lynn C. Thomason, Donald L. Court.

**Writing – original draft:** James Sawitzke, Lynn C. Thomason.

**Writing – review & editing:** James Sawitzke, Nina Costantino, Adriana Castillo Caballero, Ellen Hutchinson, Alessandro Barenghi, Lynn C. Thomason, Donald L. Court.

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
