## [Decision Letter · Decision Letter 0]

6 Jan 2026

Dear Dr. Sawitzke,

Thank you for submitting your manuscript to PLOS ONE. After careful consideration, we feel that it has merit but does not fully meet PLOS ONE’s publication criteria as it currently stands. Therefore, we invite you to submit a revised version of the manuscript that addresses the points raised during the review process.

We look forward to receiving your revised manuscript.

Kind regards,

Bashir Sajo Mienda, PhD

Academic Editor

PLOS One

Journal Requirements:

“This work was supported, in part, by the Intramural Research Program of the National Institutes of Health, National Cancer Institute, Center for Cancer Research through DLC. This project has also been partly funded with federal funds from the National Cancer Institute, National Institutes of Health, under contract no. HHSN261200800001E through DLC. Funding was provided by EMBL to J.A.S., A.C.C and A.B.”

“This work was supported, in part, by the Intramural Research Program of the National Institutes of Health, National Cancer Institute, Center for Cancer Research through DLC. This project has also been partly funded with federal funds from the National Cancer Institute, National Institutes of Health, under contract no. HHSN261200800001E through DLC. Funding was provided by EMBL to J.A.S., A.C.C and A.B.”

5. Please expand the acronym “EMBL” (as indicated in your financial disclosure) so that it states the name of your funders in full.

6. In the online submission form, you indicated that “The data are contained within the manuscript and/or Supporting information files. Data such as primer sequences are available upon request.”

7. We note that you have included the phrase “data not shown” in your manuscript. Unfortunately, this does not meet our data sharing requirements. PLOS does not permit references to inaccessible data. We require that authors provide all relevant data within the paper, Supporting Information files, or in an acceptable, public repository. Please add a citation to support this phrase or upload the data that corresponds with these findings to a stable repository (such as Figshare or Dryad) and provide and URLs, DOIs, or accession numbers that may be used to access these data. Or, if the data are not a core part of the research being presented in your study, we ask that you remove the phrase that refers to these data.

8. We notice that your supplementary [figures/tables] are included in the manuscript file. Please remove them and upload them with the file type 'Supporting Information'. Please ensure that each Supporting Information file has a legend listed in the manuscript after the references list.

9. Please update your submission to use the PLOS LaTeX template. The template and more information on our requirements for LaTeX submissions can be found at http://journals.plos.org/plosone/s/latex.

Reviewers' comments:

Reviewer's Responses to Questions

**Comments to the Author**

1. Is the manuscript technically sound, and do the data support the conclusions?

Reviewer #1: Yes

2. Has the statistical analysis been performed appropriately and rigorously?

Reviewer #1: Yes

3. Have the authors made all data underlying the findings in their manuscript fully available?

Reviewer #1: Yes

4. Is the manuscript presented in an intelligible fashion and written in standard English?

Reviewer #1: Yes

Reviewer #1: The manuscript by Sawitzke et al entitled “Enhancement of RecET-mediated in vivo linear DNA assembly by a xonA mutation” shows how a deletion of the xonA gene, which encodes a 3’ ssDNA exonuclease from E. coli, greatly stimulates the in vivo cloning capabilities following electroporation of linear DNA fragments into a strain that expresses the RecET phage recombination system. The authors first show what was already known, that the RecET system is more efficient at linear DNA fragment annealing compared to the lambda Red recombination system. They test increasing homology lengths where they show that the higher efficiency of the RecET system is most pronounced as smaller homologies (~50 bp).

Their most significant finding is the deletion of xonA results in higher efficiencies with both systems, with up to a 70-fold increase in the case of RecET acting on 2 or 3 overlapping linear fragments. Significantly, the RecET-xonA system allowed 50 bp homologies to work efficiently, allowing homologies to be added to the primers during PCR-amplification of linear DNA fragment substrates. Also, the RecET-xonA system showed the greatest effect relative to the xonA+ strain, where the recombination efficiency was increased 4 orders of magnitude when six fragments with overlapping homologies were used to assemble the recombinant plasmid. In all cases, they followed correct plasmid assembly by selecting of drug-resistant markers or the presence of blue colonies with annealed fragments due to the presence of the lacZ gene in the recombinant plasmid; the accuracy was generally 90% or greater.

Finally, the authors finally compare their in vivo system to NEB’s Gibson assembly in vitro kit and find the two systems generally compatible, suggesting that use of the RecET-xonA strain would save time without the need to treat the linear fragments with DNA modifying enzymes in vitro.

The manuscript is well written and the figures are clear. The experiments are sound and appropriate controls and statistical analyses are appropriate.

Minor points:

The only difficulty I found was that the Figures were not labelled. In particular, the last figure in manuscript entitled “DNA Fragment Assembly” is not discussed in the text. Should this figure be labelled Figure 7B?

On page 6, the reference to “Figure S3” is not correct. Fig. S3 contains lacZ sequence information from white colonies, not patching colonies on to LB-kan or LB-cam plates. Is there some other Supp Figure that shows the patching results?

**Do you want your identity to be public for this peer review?** For information about this choice, including consent withdrawal, please see our For information about this choice, including consent withdrawal, please see our Privacy Policy .

Reviewer #1: No

---

## [Author Response · Author response to Decision Letter 1]

28 Jan 2026

January 26, 2026

Dear Dr. Mienda,

Thank you for editing and sending out to review our manuscript entitled “Enhancement of RecET mediated in vivo linear DNA assembly by a xonA mutation”. We appreciate your comments and those of the reviewer and have addressed them as follows.

Editor Comments:

1., 2. We have made every effort to format the manuscript to the PLOS One guidelines.

3., 4. We have corrected our funding statement to say, “This work was supported, in part, by the Intramural Research Program of the National Institutes of Health, National Cancer Institute, Center for Cancer Research through D.L.C. This project has also been partly funded with federal funds from the National Cancer Institute, National Institutes of Health, under contract no. HHSN261200800001E through D.L.C. Funding was also provided by the European Molecular Biology Laboratory to J.A.S., A.C.C and A.B. There was no additional external funding received for this study. The funders had no role in the study design, data collection and analysis, decision to publish, or preparation of the manuscript.”

5. We have spelled out “EMBL” in that statement.

6. We have changed the statement, “Sequences of DNA oligos are available upon request” to “Sequences of DNA oligos are in S2 Table” and added S2 Table with their sequences.

7. On line 206 it used to say, “(data not shown)”. We have removed that phrase and made no further changes. The statement indicates the result as is.

8. Supporting information has been moved to separate files as requested. Also, the legends to the supporting files have been added to the main manuscript after the reference list.

10., 11. References have been checked and are accurate to our knowledge.

Reviewer Comments:

1. Figures: Checked that all references to figures are now correct. On line 143, deleted reference to “S3 Fig” as it indeed was incorrect.

Sincerely,

James A. Sawitzke, Ph.D.

jim.sawitzke@gmail.com

---

## [Editor Report · Decision Letter 1]

20 Feb 2026

Enhancement of RecET-mediated in vivo linear DNA assembly by a xonA mutation

PONE-D-25-63366R1

Dear Dr. Sawitzke,

We’re pleased to inform you that your manuscript has been judged scientifically suitable for publication and will be formally accepted for publication once it meets all outstanding technical requirements.

Kind regards,

Bashir Sajo Mienda, PhD

Academic Editor

PLOS One
---

## [Editor Report · Acceptance letter]

PONE-D-25-63366R1

PLOS One

Dear Dr. Sawitzke,

I'm pleased to inform you that your manuscript has been deemed suitable for publication in PLOS One. Congratulations! Your manuscript is now being handed over to our production team.

Kind regards,

on behalf of

Dr. Bashir Sajo Mienda

Academic Editor

PLOS One